# Antibiotic-Loaded Polymeric Barrier Membranes for Guided Bone/Tissue Regeneration: A Mini-Review

**DOI:** 10.3390/polym14040840

**Published:** 2022-02-21

**Authors:** Manuel Toledano-Osorio, Cristina Vallecillo, Marta Vallecillo-Rivas, Francisco-Javier Manzano-Moreno, Raquel Osorio

**Affiliations:** 1Faculty of Dentistry, Colegio Máximo de Cartuja s/n, University of Granada, 18071 Granada, Spain; mtoledano@ugr.es (M.T.-O.); cvallecillorivas@hotmail.com (C.V.); mvallecillo@correo.ugr.es (M.V.-R.); rosorio@ugr.es (R.O.); 2Biomedical Group (BIO277), Department of Stomatology, Facultad de Odontología, University of Granada, 18071 Granada, Spain; 3Instituto Investigación Biosanitaria ibs.GRANADA, University of Granada, C/Doctor Azpitarte 4, Planta, 18012 Granada, Spain

**Keywords:** barrier membrane, polymer, collagen, antibiotic, bone regeneration

## Abstract

Polymeric membranes are frequently used for bone regeneration in oral and periodontal surgery. Polymers provide adequate mechanical properties (i.e., Young’s modulus) to support oral function and also pose some porosity with interconnectivity to permit for cell proliferation and migration. Bacterial contamination of the membrane is an event that may lead to infection at the bone site, hindering the clinical outcomes of the regeneration procedure. Therefore, polymeric membranes have been proposed as carriers for local antibiotic therapy. A literature search was performed for papers, including peer-reviewed publications. Among the different membranes, collagen is the most employed biomaterial. Collagen membranes and expanded polytetrafluoroethylene loaded with tetracyclines, and polycaprolactone with metronidazole are the combinations that have been assayed the most. Antibiotic liberation is produced in two phases. A first burst release is sometimes followed by a sustained liberation lasting from 7 to 28 days. All tested combinations of membranes and antibiotics provoke an antibacterial effect, but most of the time, they were measured against single bacteria cultures and usually non-specific pathogenic bacteria were employed, limiting the clinical relevance of the attained results. The majority of the studies on animal models state a beneficial effect of these antibiotic functionalized membranes, but human clinical assays are scarce and controversial.

## 1. Introduction

In 1982, Nyman et al. [1] proposed the possibility of producing periodontal tissue regeneration in humans by using a barrier membrane. This barrier membrane should avoid soft tissue cell invasion of the regenerating area, maintaining the space and facilitating the periodontal ligament derived cells or bone cells to grow into the defective area. These principles have also been employed to promote guide bone regeneration at those sites where an intraoral bone defect or insufficient bone exists, mainly caused by teeth loss, trauma, tumoral pathology or infections [2]. Currently, these guided tissue regeneration and guided bone regeneration techniques are widely accepted and are often used for clinical applications [3,4]. 

These occlusive membranes must fulfill several criteria, including space maintaining capacity, mechanical properties, osteoconductivity/osteoinductivity, and biocompatibility [5,6]. Currently, it seems that natural and artificial polymers are the best candidate materials to comply with most of these prerequisites [6]. However, it should be taken into account that in many cases, periodontal guided tissue and bone regeneration are hindered due to contamination and infection of the healing site. It seems that the placement of barrier membranes at the oral cavity creates a favorable ecological niche that facilitates the growth of some periodontal pathogens [3,4].

In order to improve the barrier membrane function, the incorporation of antibacterials has been suggested to try to inhibit bacterial contamination at the surgical intervention or during the healing period if membrane exposure to the oral cavity occurs (Figure 1) [4,7]. If bacterial colonization and subsequent infection is produced in the early stages of wound healing, the clinical outcomes of the complete procedure will be jeopardized [7,8]. Controlling the membrane’s colonization of bacteria and reducing the possibility of infection in the early healing stage increases the predictability of the clinical outcomes [9]. It should be taken into account that some issues discourage the use of systemic antibiotic therapy due to risk of toxicity, acquired bacterial resistance, difficulty in penetrating some areas, and insufficient concentration levels at the infected site to efficiently inhibit the target microorganisms, among others [10]. Therefore, the use of local drug administration is recommended to potentially reduce the drug resistance of the bacteria by lowering the dosage of used antibiotics. The combination of polymeric barrier membranes and antibacterials are preferred in order to facilitate, accelerate, and enhance the effect of guided tissue and guided bone regeneration procedures [4,7].

The estimated healing period in bone regeneration is more than 6 months, and for periodontal regeneration, 4 to 6 weeks are necessary [11]. Antibacterials have shorter lifespans and rapid local clearances at bone healing sites. To overcome these points, a polymeric carrier system may play a key role in determining antibacterial activity. In recent years, there has been a strong increase in research focused on appropriate antibacterials and carrying materials for controlled and optimal release. Polymeric-based membranes have been proposed as key biomaterials capable of securing sustained release of antibacterials over a period of time and of affording acceptable release kinetics [4].

The purpose of this study was to review the existing literature on the main findings on antibiotic-loaded polymeric barrier membranes, covering design, manufacturing, loading and release kinetic, antibacterial efficacy, and usefulness for guided bone and tissue regeneration.

## 2. Methods

Using the National Library of Medicine (MEDLINE by PubMed), The Cochrane Oral Health Group Trials Register, EMBASE, and Web of Science (WOS) a literature search was performed for papers, including peer-reviewed publications from 1963 up to January 2022. 

Combinations of several search terms were applied to create a search strategy including the following word combinations: (“Guided Tissue Regeneration” OR “GTR” OR “Guided Bone Regeneration” OR “GBR” OR “Bone Regeneration” OR “Periodontal Regeneration” OR “Bone Tissue Regeneration” OR “Bone formation” OR “Osteogenesis” OR “Osteogenic regeneration”) AND (“Barrier Membrane” OR “Membrane” OR “Barrier” OR “Collagen Membrane” OR “Chitosan-Collagen Membrane” OR “Natural Membrane” OR “Bovine Membrane” OR “Porcine Membrane” OR “Pericardium Membrane” OR “Dermis Membrane”) AND ((ions[MeSH Terms]) OR antibiotics[MeSH Terms] OR (antibacterial agents[MeSH Terms]) OR (agents, antimicrobial[MeSH Terms]) OR tetracycline OR doxycycline OR metronidazole OR minocycline OR roxithromycin OR moxifloxacin OR ciprofloxacin)). Bibliographies of eligible articles were also manually searched for missing papers after the electronic searching. 

## 3. Results and Discussion

### 3.1. Polymeric Materials for Antibacterial-Loaded Membranes

Several polymers have been used as antibacterial carriers for bone regeneration barrier membranes. They can roughly be classified as natural or synthetic, resorbable or non-resorbable [3,6]. The previously-employed polymers for the mentioned medical application are presented in Table 1.

Among all the polymeric biomaterials, the natural collagen membrane is the most widely used as an antibacterial carrier in bone regeneration [5,12,13,14,15,16,17,18,19,20,21]. Several factors may explain this finding: (1) among the degradable membranes, collagen-based ones are the most commonly employed in dentistry because of their bioactivity, biocompatibility, and mechanical properties [3,7,22]; (2) they have been found to have many options for loading [23]. It should be considered that the chemical structure of collagen offers versatility, as it contains carboxyl and amino terminals, permitting not only adsorption, but also covalent binding of a great variety of different chemical groups [24]; (3) collagen degradability permits effective antibacterial liberation even if the antimicrobial substance is covalently linked to collagen [23,24]. 

Other synthetic polymers that have also been used as antibacterial carriers are: poly(lactic acid) (PLA) [25,26,27,28], poly(glycolic acid) (PGA) [16,19,29,30,31,32], polycaprolactone (PCL) [33,34,35,36,37,38], or combinations between them [10,39,40,41,42,43]. All these abovementioned polymers are also resorbable. PCL is a slow resorbing polymer as it degrades via an erosion mechanism, hence avoiding the rapid release of acidic byproducts, which may be detrimental to surrounding tissues. PGA and PLA are aliphatic polyesters with a fast degrading behavior [29]. 

Among the non-resorbable polymers, two are used as antibacterial carriers. One of them is expanded polytetrafluoroethylene (ePTFE) [8,16,19,44,45] and the second is a novel polymer based on hydroxyethyl methacrylate–methyl methacrylate (HEMA–MMA) copolymers that is still in the experimental phase [46,47].

### 3.2. Manufacturing Procedure for Polymeric Antibacterial-Loaded Membranes

The most frequently employed manufacturing technique for synthetic polymeric membranes is electrospinning [5,25,29,30,31,33,34,35,36,37,38,41,42,47,48,51,52,60,61]. This production method permits the adjusting of the most relevant characteristics of the manufactured membranes. It enables the creation of membranes with desired mechanical properties such as flexibility or elasticity. Fiber diameter may also be adjusted from micro to nanosized. Pore size, which imparts occlusive properties and an interconnected porous network resembling the bone collagen network, which is favorable for long term tissue infiltration and integration [29,47,48], can be controlled. Processing variables for each electrospinning method are different between the evaluated studies and include different voltages, needle to collector distances, and flow rates. These variables, together with the polymeric solution parameters such as surface tension, viscosity, and electrical conductivity of the solution, control the morphology of the electrospun fiber mats [5]. 

Antibacterials can be loaded in the electrospun nanofiber through: (1) blending, which is a passive loading of the antibacterial into the nanofibers (adding it in the polymeric solution prior to electrospinning) [25,30,31,33,34,36,37,41,51]; (2) coaxial electrospinning, where the antibacterials are embedded inside the electrospun nanofibers in order to improve some different aspects such as release outline (extending the period of drug delivery), drug safety, or drug-loading efficiency of non-soluble substances [37,38,42,55]; and (3) solvent evaporation or immersion techniques after fiber production, which permit physical absorption and chemical bonding of the antibacterials onto the polymers [37,47,48]. The simple electrospinning technique has gained widespread interest in the area of tissue engineering and drug delivery due to its relative ease of use and versatility [62]. Meanwhile, co-axial electrospinning is less employed, as it is a more difficult technique requiring more than a single nozzle [5]. One of the major advantages of electrospun fiber mats is the inherently high surface-to-volume-ratio of formed scaffolds. Not only does this help to enhance drug loading and to accomplish sustained and controlled local drug delivery, but it also improves cell attachment [62]. 

Collagen, PLA, PLGA, PCL, and other polymeric resorbable membranes have been fabricated through the casting method, by solvent evaporation, or as dried films [10,12,20,39,43,50,55]. In these cases, they are less porous and do not have a fibrous micro or nanostructure resembling collagen. These membranes were loaded by incorporation of the antibacterial in the polymer blend solution [10,12,13,32,39,49,51,55] or by immersion or solvent evaporation techniques after membrane production [14,15,16,17,18,19,20,43,50]. 

When using non-resorbable synthetic membranes, antibacterials are coated on the outer polymer surface through adsorption [8,16,44,59], direct covalent binding of the drug onto the membrane surface [47,48], or by grafting (using intermediary compounds in order to provoke a crosslinking reaction between the antibacterial and the polymeric membrane) [59]. 

Other manufacturing techniques as supercritical CO_2_-assisted processes, 3D printing, porogen leaching, gas foaming, phase separation, or any possible combination between these may also be employed for polymeric membrane preparation [63,64]. Among these techniques, the phase separation process is easy to execute and does not require sophisticated equipment. It is based on the principle that a homogeneous solution of a polymer dissolved in a good solvent can undergo a phase separation, causing solution saturation that will lead to polymer precipitation, followed by a microcellular foam polymer structuration [65]. It is beneficial since it may offer good control of the scaffold structure, particularly in terms of porosity and interconnectivity [63]. These properties play a significant role in tissue regeneration, affecting several cell processes such as adhesion, migration, proliferation, and differentiation [65].

### 3.3. Loaded Antibacterials for Bone Regeneration in Dentistry

The most frequently loaded antibiotics in bone regeneration membranes are tetracyclines [8,12,13,14,16,17,18,27,28,29,30,32,39,40,43,47,48,49,50,51,56,57,59]; which are broad-spectrum antibiotics that have been shown to be useful in fighting against most of the bacteria responsible for periodontitis [66]. The most frequently employed tetracyclines have been minocycline [12,13,49] and doxycycline [14,28,30,47,48,50,56,59]. Tetracyclines work by inhibiting protein synthesis in bacteria [67] and have been shown to have a prolonged lifespan and anticollagenase properties, and are well absorbed by bone due to a calcium quelating effect [59]. 

The second most frequently employed antibiotics are metronidazole [5,10,15,20,21,25,33,38,42,52,55,58] and other nitroimidazoles such as ornidazole [31], niridazole, and tinidazole [21]. These are antibiotics with antibacterial activity for Gram-negative and anaerobic bacteria [25], and they are specific against most of the subgingival [10] and periodontopathic biofilms [21]. Metronidazole’s mechanism of action is based on the alteration of nucleic acid synthesis in bacteria [67].

Other encountered, but less used antibiotics for loading membranes were amoxycillin [15,16,19], vancomycin [41] or azithromycin [29]. Amoxycillin and vancomycin are antibiotics targeting the bacterial cell wall [67]. Azithromycin is a macrolide antibiotic extensively recommended for a wide range of anaerobic infections. It mainly acts by altering protein synthesis. However, its main disadvantage is low bio-availability as a result of its poor water solubility, probably limiting its proposed clinical application [29]. 

### 3.4. Antibiotic Release Kinetics

Antibiotic release kinetics is not evaluated in all the reviewed studies. When ascertained, it was usually done in vitro. The supernatants are measured at specific time-points after immersion of the loaded membranes in a solution (normally deionized water or phosphate buffered saline). High performance liquid chromatography [31,33,34,37,38,41], inductively coupled plasma atomic emission spectrometry [53], UV-vis spectrophotometry [10,12,13,21,30,50,58,59], and fluorescence [36] are the most used techniques to determine the released concentration of the loaded antibiotics. Loading efficiency, when determined, was usually high, ranging between 30 and 85% [29,52].

Antibiotic release from polymeric membranes is in all cases characterized by two different phases. The first is an initial burst release, which can be described as the liberation produced between 7 and 10 h [37,55,68], or between 12 and 48 h [10,13,29,31,32,33,34,36,39,42,49,50,52,54,58,59]. This rapid release is followed by a slower and sustained liberation that may last from 35 h to 10 days [10,13,21,32,34,37,38,39,48,52,57,59,65], or in some cases much longer, up to 28 days [12,25,30,33,36,49,53,58]. These described liberation kinetics indicate that some of the antibiotic is always retained by absorption and is rapidly liberated after immersion. The slow and relatively sustained posterior release probably corresponds to the antibiotic that is ionically or covalently linked to the membranes, or to the antibiotic that is liberated at the same time that the membrane degrades. Therefore, the procedures used to load the antibiotic and the type of membrane may be considered as determinant factors affecting antibiotic release. The loading of antibiotics on polymers through chemical conjugation may have a more controlled kinetic release than those processed through physical adsorption [69]. However, it is also necessary to investigate the antibacterial activity of the released antibiotics, since liberation does not always imply biological activity. 

### 3.5. Antibacterial Efficacy of Antibiotic-Loaded Membranes

Most studies investigated antibiotic loaded membrane efficacy using in vitro antibacterial cell assays [12,15,16,19,29,30,31,33,34,38,41,42,44,48,49,53,56,58,65] and/or in vivo animal models when surgically treating contaminated bone defects [14,36,41]. The efficacy of the membrane as an antibiotic delivery carrier was always confirmed. 

Of the antibacterial cell assays, the most employed was the agar disk diffusion test [12,15,29,30,31,33,34,38,41,42,49,56,58,65]. Other techniques such as the plate-counting method [53], determination of bacterial penetration through membranes [19], bacterial colonization on membranes [44], scanning electron microscopy evaluation of membranes colonization [16,48,53], or number of cells determination by more precise techniques such as quantitative polymerase chain reaction [48] were rarely executed.

It is also worth mentioning that sometimes non-clinically relevant or unspecific bacteria were used for these studies, as in the case of *Peptostreptococcus anaerobius* [31], *Staphylococcus aureus* [12,30,33,36,41,53,56], *Pseudomonas aeruginosa* [33], *Echerichia coli* [30,53,56], *Helicobacter pilori* [37], or *Streptococcus mutans* [16,19,49]. Of periodontally-relevant bacteria *Aggregatibacter actinomycetemcomitans* [15,16], *Porphyromonas gingivalis* [13,14,15,49] and *Fusobacterium nucleatum* [13,31,34,38,42,68] have been tested. All these studies were based on single bacterial cultures; therefore, results should not be directly extrapolated to the clinical situation. It has to be taken into account that bacteria grow in biofilms, providing them with specific characteristics that make bacteria more resistant and tolerant to antibiotics than when in a planktonic state [70]. Only one recently published study was performed using a subgingival multispecies biofilm model with six different bacterial species [48].

The incorporation of antibacterial agents in membranes is a promising approach that may promote bone formation, especially for some challenging clinical situations when the characteristics of the defect make the site especially prone to membrane exposure and subsequent bacterial contamination and infection. However, despite the promising results encountered in vitro and in preclinical animal models, the value of incorporating antibacterials has not yet been evidenced clinically [4]. 

In animal models, when treating previously contaminated bone defects, the efficacy has been probed in terms of bone regeneration of doxycycline-loaded collagen membranes [14], vancomycin-loaded PCL-membranes [36], and vancomycin-loaded PGA membranes [41]. Tetracycline-loaded PGA membranes and minocycline-chitosan membranes also induced major regeneration in periodontal defects in beagle dogs [51] and rats [49], respectively.

None of the antibacterial-loaded membranes have been evaluated for efficacy in reducing microbial adhesion and infection in humans. Conversely, several antibiotic-loaded membranes were tested in humans evaluating clinical efficacy when compared to non-antibiotic-loaded membranes. Gain in periodontal attachment level and increases in bone formation were obtained when using tetracycline-loaded ePTFE membranes [44], doxycycline-loaded collagen membranes [18], and metronidazole-loaded collagen membranes [20]. However, in two clinical studies, doxycycline-loaded collagen membranes [28] and tetracycline-loaded ePTFE membranes [8] failed to enhance the periodontal regeneration outcomes when compared to non-antibiotic-loaded membranes. These controversial results may be due to the small sample size of the studies (around 10 to 25 patients) and to the lack of standardization of the employed antibiotic concentration and liberation, which was sometimes extremely low (i.e., 4 wt% [28]) or not reported [8]. 

### 3.6. Other Findings Associated with Antibiotics Loaded on Polymeric Membranes

The cytocompatibility of these membranes was sometimes evaluated using different cells lines as osteoblasts [13,37,42,53,60,61], fibroblasts [13,35,37,39,52], epithelial cells [59], and stem cells [5,25,30,32], always with favorable results. 

It is relevant that in addition to their antimicrobial activity, doxycycline and minocycline have been shown to enhance osteoblast and/or stem cell proliferation, differentiation, and osteogenic activity [5,13,30,32,33,53,60,61]. Moreover, these antibiotics were shown to inhibit bone resorption and to promote bone formation when assayed in animals [13,29,33,40,43] and in humans [62].

Immunomodulatory effects have been proven for doxycycline-loaded membranes in both cells [71] and animals [47]. The same effect was also shown for metronidazole-loaded collagen membranes in cells [21] or azithromycin-loaded PGA membranes when tested in an animal model [29].

Tetracycline-loaded collagen membranes have also been reported to have slower degradation [17], which may be beneficial for bone regeneration in challenging bone defects. 

## 4. Conclusions

It can be concluded that, taking into account the fact that infection can lead to the failure of the intended bone regeneration, polymeric membranes could be used as carriers for local antibiotic therapy. Due to antibiotic lifespans and the rapid clearance rate existing at the surgical sites, it is impossible for antibiotics to produce a long-term effect without the aid of a carrier facilitating a controlled liberation. The loading efficacy and the kinetic release will depend on the employed polymeric material. The polymeric carrier should ideally have a constant and slow degradation and should be ideally maintained through the complete healing period. Collagen or ePTFE loaded with tetracyclines, and PCL with metronidazole are the most frequently assayed combinations. Antibiotics present the advantage of possessing a wide therapeutic window, making it easier to obtain a beneficial effect whenever the liberation is effective. In the existing studies, even when antibacterial efficacy is often reported, most of the times it is not measured against specific subgingival pathogenic bacteria and it is usually measured using an agar disc diffusion method, which are two variables limiting the clinical relevance of the previously published results. It should be considered that the present literature review lacks of standardization in method; therefore, results need to be taken with caution.

It should also be taken into account that there is relatively scarce experimental evidence that a local antibacterial strategy could be useful in bone regeneration procedures. Apart from several studies on infected periodontal defects locally treated with antibiotics and polymeric membranes [8,18,20,28,44], no specific antimicrobial strategy has been yet clinically validated [4]. 

Future studies should be performed focusing on: (i) the standardization of adsorption/release abilities of the different polymeric carriers, (ii) antibacterial activity assays using specific and periodontal clinically-relevant biofilm models, and (iii) randomized clinical trials in order to finally determine the safety and efficacy of these novel and innovative procedures; thereby helping to eliminate the barriers limiting the extension of the experimental results to the clinical situation.

## Figures and Tables

**Figure 1 polymers-14-00840-f001:**
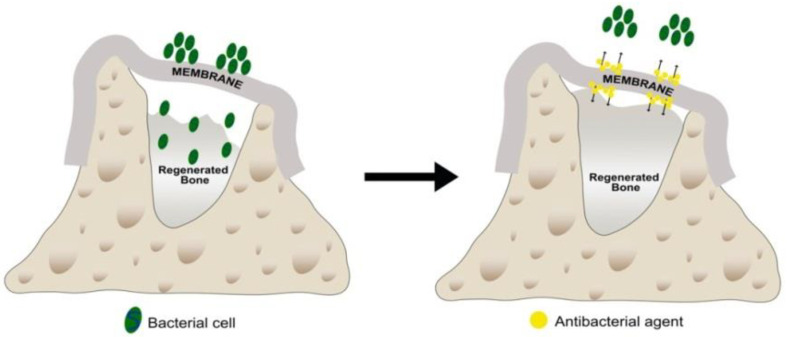
A barrier membrane employed to avoid soft tissue cell invasion, enhancing space maintenance of the regenerating area. The incorporation of antibacterials in the membrane has been suggested to inhibit bacterial contamination during the surgical intervention or the healing period if membrane exposure to the oral cavity occurs, improving the performance of the bone regeneration procedure.

**Table 1 polymers-14-00840-t001:** Combinations of previously-employed polymers and antibiotics in the designing of barrier membranes for guided bone/tissue regeneration.

Polymeric Material	Origin	Resorbable	Loaded Antibiotic	References
Expanded Polytetrafluoroethylene-ePTFE-	Synthetic	No	Tetracycline	[8,16,19,44]
			Amoxicillin	[16,19]
(MMA)1-co-(HEMA)1/(MA)3-co-(HEA)2	Synthetic	No	Doxycycline	[47,48]
Collagen	Natural or Synthetic	Yes	Minocycline	[12]
			Doxycycline	[14]
			Tetracycline	[16,17,18,19]
			Amoxicillin	[15,16,19]
			Metronidazole	[15,20,21]
			Niridazole	[21]
			Tinidazole	[21]
Chitosan		Yes	Minocycline	[49]
			Doxycycline	[50]
Collagen-Chitosan		Yes	Minocycline	[13]
Poly(lactic acid) -PLA-	Synthetic	Yes	Metronidazole	[25]
			Doxycycline	[28]
			Tetracycline	[27]
Poly(glycolic acid) -PGA-	Synthetic	Yes	Azithromycin	[29]
			Doxycycline	[30]
			Tetracycline	[16,19,51]
			Amoxicillin	[16,19]
			Ornidazole	[31]
Polycaprolactone-PCL-	Synthetic	Yes	Moxifloxacin	[33]
			Metronidazole	[34,37,38,52]
			Vancomycin	[36]
			Salicylic acid	[35]
PGA-PLA	Synthetic	Yes	Tetracycline	[39,40,43]
			Vancomycin	[41]
			Metronidazole	[10,42]
Polyetheretherketone	Synthetic	No	Gentamicin	[53]
Hyaluronic acid	Synthetic	Yes	Hinokitiol	[54]
			Metronidazole	[55]
Cellulose	Synthetic	Yes	Doxycycline	[56]
			Tetracycline	[57]
Hydroxybutyrate	Synthetic	Yes	Metronidazole	[58]
Silk fibroin	Synthetic	Yes	Tetracycline	[32]
Polyvinylidene difluoride-PVDF-	Synthetic	No	Doxycycline	[59]

## Data Availability

The data presented in this study are available on request from the corresponding author.

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
