# Peer review of "Antibiotic-Loaded Polymeric Barrier Membranes for Guided Bone/Tissue Regeneration: A Mini-Review"

_polymers, 2022, doi:10.3390/polym14040840_

Round 1

Reviewer 1 Report

The manuscript “Antibiotic-loaded polymeric barrier membranes for guided bone/tissue regeneration” is a review on the production of polymeric membranes to be used as carriers for local antibiotic therapy. The work is clear and well organized. However, some revisions are required, as follows:

- The title should be modified, adding “Mini Review”.

- Abstract. Add mechanical and morphological properties of the membranes, required for this application; e.g., porosity value, pore size, mechanical resistance, Young modulus, etc..

- Avoid the use of Materials and Methods paragraph.

- Among the preparation methods of the membranes, phase separation and supercritical assisted phase separation have to be described, since they are the processes most frequently used. For this purpose, see for instance these works: Baldino te al., Production, characterization and testing of antibacterial PVA membranes loaded with HA-Ag3PO4 nanoparticles, produced by SC-CO2 phase inversion, Journal of Chemical Technology and Biotechnology, 2019, 94(1), pp. 98–108;

Prihandana et al., Study Effect of nAg Particle Size on the Properties and Antibacterial Characteristics of Polysulfone Membranes, Nanomaterials, 2022, 12(3), 388; etc.

- The last paragraph could be entitled: Discussion and Conclusions.

Author Response

REVIEWER 1:

Comments and Suggestions for Authors

Reviewer: The manuscript “Antibiotic-loaded polymeric barrier membranes for guided bone/tissue regeneration” is a review on the production of polymeric membranes to be used as carriers for local antibiotic therapy. The work is clear and well organized. However, some revisions are required, as follows:

R#: The title should be modified, adding “Mini Review”.

Our response: We completely agree with the reviewer. It has been done. The title now reads:

…’Antibiotic-loaded polymeric barrier membranes for guided bone/tissue regeneration: a mini-review’…

R#: Abstract. Add mechanical and morphological properties of the membranes, required for this application; e.g., porosity value, pore size, mechanical resistance, Young modulus, etc..

Our response: The reviewer is right, this point was missing. The following sentence was added:

…’Polymers provide adequate mechanical properties (i.e. Young modulus) to support oral function and also pose some porosity with interconnectivity to permit for cells proliferation and migration.’…

R#: Avoid the use of Materials and Methods paragraph.

Our response: The reviewer is right. The section ‘Materials and Methods’ has been deleted. Instead, as it is necessary to concisely describe reproducible methods for conducting the review (databases, access dates and key words), just a reduced ‘Methods’ paragraph was maintained.

R#: Among the preparation methods of the membranes, phase separation and supercritical assisted phase separation have to be described, since they are the processes most frequently used. For this purpose, see for instance these works:

Our response: The reviewer is completely right. Mentioned techniques have been included at the end of the section informing about membranes manufacturing. Proposed references have also been included. The new paragraph reads:

… Other manufacturing techniques as supercritical CO2 assisted processes, 3D printing, porogen leaching, gas foaming, phase separation or even any possible combination between them may also be employed for these polymeric membranes preparation [63, 64]. Among these techniques, the phase separation process is easy to execute and does not require a sophisticated equipment. It is based on the principle that a homogeneous solution of a polymer dissolved in a good solvent can undergo a phase separation, causing the solution saturation which will lead to polymer precipitation, followed by a microcellular foam polymer structuration [65]. It is beneficial as it may offer a good control of the scaffold structure, particularly porosity and interconnectivity [63]. These properties play a significant role in tissue regeneration, affecting on several cell processes like adhesion, migration, proliferation and differentiation [65]

  1. Baldino et al., Production, characterization and testing of antibacterial PVA membranes loaded with HA-Ag3PO4 nanoparticles, produced by SC-CO2 phase inversion, Journal of Chemical Technology and Biotechnology, 2019, 94(1), pp. 98–108

  1. S. Gay, G. Lefebvre, M. Bonnin, B. Nottelet, F. Boury, A. Gibaud, B. Calvignac, PLA scaffolds production from Thermally Induced Phase Separation: Effect of process parameters and development of an environmentally improved route assisted by supercritical carbon dioxide, The Journal of Supercritical Fluids,Volume 136, 2018,Pages 123-135. https://doi.org/10.1016/j.supflu.2018.02.015

  1. Prihandana et al., Study Effect of nAg Particle Size on the Properties and Antibacterial Characteristics of Polysulfone Membranes, Nanomaterials, 2022, 12(3), 388.

R#:  The last paragraph could be entitled: Discussion and Conclusions.

Our response: It is an interesting approach. However, we have taken into account that most of the Discussion section is incorporated at the third part of the manuscript (3. Results and Discussion). Therefore, we have considered entitling the last paragraph as ‘Conclusions’. Otherwise, we have tried not to make changes at the manuscript headings, as the second reviewer included an specific comment about employed structure and sections, and literally said: ‘Headings are consistent and fine’.

Reviewer 2 Report

Review article titled (Antibiotic-loaded polymeric barrier membranes for guided bone/tissue regeneration) by Toledano-Osorio et al. discussed the methods of manufacturing and usefulness of Antibiotic-loaded polymeric barrier membranes. I appreciate such simple and humble review for discussing a clinically important topic. I have some suggestions for the overall presentation of the topic:

1-The aim at the end of the introduction needs reformulation to cover all the topics, for example "methods of preparations" are discussed in the review but not mentioned in the aim...etc

2-Headings are  consistent and fine

3- I noticed many times in the review that multiple references (up to 10) appear together. Please divide them into parts and talk about on every 3-4 references max together instead of packaging high number of papers together.

Author Response

REVIEWER 2

Comments and Suggestions for Authors

Reviewer: Review article titled (Antibiotic-loaded polymeric barrier membranes for guided bone/tissue regeneration) by Toledano-Osorio et al. discussed the methods of manufacturing and usefulness of Antibiotic-loaded polymeric barrier membranes. I appreciate such simple and humble review for discussing a clinically important topic. I have some suggestions for the overall presentation of the topic:

R#: 1-The aim at the end of the introduction needs reformulation to cover all the topics, for example "methods of preparations" are discussed in the review but not mentioned in the aim...etc

Our response: The reviewer is completely right. It has been completely re-written and it now reads as follows:

…’The purpose of this study was to review the existing literature about main findings on antibiotic–loaded polymeric barrier membranes; covering design, manufacturing, loading and release kinetic, antibacterial efficacy and usefulness for guided bone and tissue regeneration.’…

R#: 2-Headings are consistent and fine

Our response: We really appreciate the comment.

R#: 3- I noticed many times in the review that multiple references (up to 10) appear together. Please divide them into parts and talk about on every 3-4 references max together instead of packaging high number of papers together.

Our response: We have tried very hard to adapt to the reviewer’s consideration. In most of the cases, it has been possible, and references have not been grouped together, and were divided into parts. However, in some cases, we are citing all the manuscripts that accomplished for a specific condition (e.g., using tetracyclines, or employing collagen-based membranes) and there is no other possibility to do it, than putting them together. 

Round 2

Reviewer 2 Report

The revised form of review article titled (Antibiotic-loaded polymeric barrier membranes for guided bone/tissue regeneration: a mini-review) by Toledano-Osorio et al. was improved according to the reviewer's recommendation & I recommend this revised form for publication in Polymers.